# Chitosan Oligosaccharide Attenuates Nonalcoholic Fatty Liver Disease Induced by High Fat Diet through Reducing Lipid Accumulation, Inflammation and Oxidative Stress in C57BL/6 Mice

**DOI:** 10.3390/md17110645

**Published:** 2019-11-16

**Authors:** Wenjing Tao, Wanjing Sun, Lujie Liu, Geng Wang, Zhiping Xiao, Xun Pei, Minqi Wang

**Affiliations:** Key Laboratory of Molecular Animal Nutrition, Ministry of Education, College of Animal Science, Zhejiang University, Hangzhou 310058, China; taowenjing1127@163.com (W.T.); sunwanjing@zju.edu.cn (W.S.); liulj@zju.edu.cn (L.L.); wanggeng@zju.edu.cn (G.W.); xiaozp@zju.edu.cn (Z.X.); pxpeixun@zju.edu.cn (X.P.)

**Keywords:** chitosan oligosaccharide, nonalcoholic fatty liver disease, lipid accumulation, inflammation, oxidative stress, AMPK

## Abstract

Nonalcoholic fatty liver disease (NAFLD) has become the most common chronic liver disease closely associated with metabolic syndrome, but there are no validated pharmacological therapies. The aim of this study was to investigate the effect of chitosan oligosaccharide (COS) on NAFLD. Mice were fed either a control diet or a high-fat diet (HFD) with or without COS (200 or 400 mg/kg body weight (BW)) by oral gavage for seven weeks. Administration with COS significantly lowered serum lipid levels in the HFD-fed mice. The hepatic lipid accumulation was significantly decreased by COS, which was attributed to decreased expressions of lipogenic genes and increased expressions of fatty β-oxidation-related genes. Moreover, pro-inflammatory cytokines, neutrophils infiltration, and macrophage polarization were decreased by COS in the liver. Furthermore, COS ameliorated hepatic oxidative stress by activating the nuclear factor E2-related factor 2 (Nrf2) pathway and upregulating gene expressions of antioxidant enzymes. These beneficial effects were mediated by the activation of the adenosine monophosphate-activated protein kinase (AMPK) signaling pathway. Therefore, COS might be a potent dietary supplement to ameliorate NAFLD.

## 1. Introduction

Nonalcoholic fatty liver disease (NAFLD), characterized by excessive fat accumulation in hepatocytes, has become the most common chronic liver disease, and its prevalence is up to one-fourth of the adults worldwide [1,2]. This disease is closely associated with metabolic syndrome, such as obesity, hyperlipidemia, hypertension, and diabetes mellitus [3]. It has been demonstrated that patients with metabolic syndrome have a higher prevalence of NAFLD than healthy people, and the incidence rates of metabolic syndrome were significantly higher in NAFLD patients than that in normal individuals [4]. In order to elucidate the pathogenesis of NAFLD and find out the effective treatment strategies, a high-fat diet (HFD) is widely used to induce NAFLD in experimental animals [5,6]. Previous studies demonstrated that HFD induced hepatic fat accumulation by accelerating lipogenesis and depressing fatty acid oxidation [5,7]. A transcriptome analysis demonstrated that HFD upregulated pro-inflammatory genes and downregulated the genes related to antioxidative capacity [8]. These findings suggested that HFD-induced NAFLD is linked to the disorder of lipid metabolism, inflammation, and oxidative stress in the liver. Therefore, food supplements, which possess anti-inflammatory and/or antioxidative effects, may be beneficial for prevention and treatment of HFD-induced NAFLD. For instance, green tea has anti-inflammatory and antioxidative activities and has been demonstrated to mitigate HFD-induced NAFLD in different rodent models [9]. Vitamin E as an antioxidant was reported to treat NAFLD [10].

As the second most abundant polymer in nature, chitin is mainly found in the shells of crustaceans, such as crab and shrimp [11]. Chitosan oligosaccharide (COS) is a degradation product of chitin via deacetylation and hydrolysis [12]. Numerous studies have shown that COS could suppress the inflammatory responses in various types of cells and animal models, and it possesses strong radical scavenging and antioxidative capacities [13,14]. In addition, COS has been demonstrated to ameliorate HFD-induced body weight gain and reduce serum and hepatic lipid profile in mice [15]. Zheng et al. reported that COS decreased the blood glucose, reversed the insulin resistance and suppressed the inflammatory responses in adipose tissue of diabetic mice [16]. These studies have demonstrated the beneficial effects of COS on metabolic disorders, including obesity, dyslipidemia, hyperglycemia, and diabetes mellitus. Bai et al. reported that COS improved hepatic glucolipid metabolism disorder by suppressing inflammation response in HFD-fed mice [17]. However, there has been limited study focusing on the therapeutic effect of COS on NAFLD, as well as the underlying mechanisms. Whether the effect is linked to the anti-inflammatory and antioxidative capacities of COS has not been clarified. Therefore, the purpose of this study was to investigate the effect of COS on HFD-induced NAFLD in C57BL/6 mice and to explore the potential mechanisms involved.

## 2. Results

### 2.1. Effects of COS on Serum Biochemical Parameters

The effects of COS on the serum lipid profile are presented in Table 1. Compared to the normal chow diet (NCD) group, the mice fed HFD had significantly higher serum levels of total cholesterol (TC), high-density lipoprotein (HDL), and low-density lipoprotein (LDL), but there was no significantly difference in triacylglycerol (TG) level among all the groups. Intragastric administration with 400 mg/kg body weight (BW) COS led to significantly lower serum levels of TC and LDL than those in the HFD group. The LDL/HDL ratio, elevated by HFD, was reduced by both dosages of COS administration (200 and 400 mg/kg BW). In addition, the serum aspartate aminotransferase (AST) and alanine aminotransferase (ALT) levels in the HFD group were significantly higher than those in the NCD group, whereas the administration with 400 mg/kg BW COS significantly decreased the serum AST and ALT levels (Table 1).

### 2.2. Effects of COS on Hepatic Steatosis

The liver weight was not changed by HFD (Figure 1A), but the hepatic TC and TG contents were significantly higher compared with those in the NCD group (Figure 1B,C). The mice administrated with COS (200 or 400 mg/kg BW) had significantly decreased the liver weight and the hepatic TC and TG contents, compared to the mice in the HFD group. Hematoxylin and eosin (H&E), and Oil Red O staining of the liver tissue also demonstrated the protective effects of COS against HFD-induced hepatic lipid accumulation. As shown in Figure 1D,E, compared to those in the NCD group, lipid vacuoles and lipid droplets were increased in the HFD group, whereas in the HFD + low dosage of COS (HFDLC) and HFD + high dosage of COS (HFDHC) groups, lipid accumulation were reduced by alleviating these histological alterations.

The mRNA expressions of genes related to the hepatic lipid metabolism were analyzed to elucidate the underlying mechanisms of COS-mediated reduced hepatic lipid accumulation. As shown in Figure 2, the mRNA levels of lipogenic genes sterol regulatory element-binding protein-1c (*SREBP-1c*) and fatty acid synthase (*FAS*) were upregulated by HFD, whereas both dosages of COS administration (200 and 400 mg/kg BW) downregulated the *SREBP-1c* mRNA expression, and the administration with 400 mg/kg COS decreased the *FAS* mRNA level. Additionally, the mRNA expressions of fatty β-oxidation-related genes, including peroxisome proliferator-activated receptor alpha (*PPARα*) and carnitine palmitoyltransferase 1 (*CPT-1*), were significantly higher in the HFDHC group than those in the HFD group.

### 2.3. Effects of COS on Hepatic Inflammation Response

As shown in Figure 3A–C, the levels of pro-inflammatory cytokines tumor necrosis factor-α (TNF-α), interleukin-1β (IL-1β), and interleukin-6 (IL-6) in the livers of HFD-fed mice were significantly increased compared with those in the mice fed with NCD, whereas the hepatic TNF-α level was decreased in the HFDHC group, and the IL-6 levels were lowered both in the HFDLC and HFDHC groups. The COS administration (200 and 400 mg/kg BW) also significantly reversed the elevated hepatic myeloperoxidase (MPO) activity induced by HFD (Figure 3D). Moreover, increased mRNA levels of *F4/80* and *CD11c* and elevated F4/80 expression detected by immunohistochemical staining were observed in the livers of HFD group compared with those in the NCD group (Figure 3E–G). However, these effects were markedly alleviated by the COS administration (200 or 400 mg/kg BW).

### 2.4. Effect of COS on Hepatic Oxidant Stress

As shown in Figure 4, there were no significantly differences in total antioxidant capacity (T-AOC) and catalase (CAT) activities among all the groups. However, the mice in the HFD group showed significantly lower activities of hepatic glutathione peroxidase (GSH-Px) and higher levels of malonaldehyde (MDA) and nitric oxide (NO) compared to those in the NCD group. Compared with those in the HFD group, hepatic GSH-Px and superoxide dismutase (SOD) activities were significantly increased in the HFDHC group, while MDA and NO levels were significantly decreased in both the HFDLC and HFDHC groups.

Subsequently, the mRNA expressions of key molecules related to oxidative stress were measured. As shown in Figure 5, the mRNA levels of NADPH quinone oxidoreductase 1 (*NQO1*), heme oxygenase-1 (*HO-1*), and glutathione S-transferase alpha 1 (*GSTA1*) were decreased in the HFD group, compared with those in the NCD group. However, the *NQO1* mRNA expression was markedly upregulated by the COS administration (200 or 400 mg/kg BW), while the mRNA levels of *HO-1* and *GSTA1* were significantly increased by the administration with 400 mg/kg COS.

### 2.5. Effects of COS on Hepatic Protein Expression

To explore the mechanisms involved in the hepatoprotective effect of COS, we examined the protein expression of nuclear factor E2-related factor 2 (Nrf2) and the phosphorylated levels of adenosine monophosphate-activated protein kinase (AMPK) and acetyl-CoA carboxylase (ACC). As shown in Figure 6, the Nrf2 expression was lower in the livers of the HFD group than that in the NCD group, whereas 400 mg/kg BW COS significantly reversed this effect. We also found the decreased phosphorylated levels of AMPK and ACC induced by HFD was upregulated by 400 mg/kg BW COS.

### 2.6. Correlation between the Studied Variables

To elucidate the relationship among the serum lipid profile, the hepatic lipid accumulation, inflammation, the oxidative stress, and the related signaling pathway, the Spearman’s correlation coefficients were calculated, as shown in Figure 7. The serum lipid level and the hepatic lipid accumulation were positively correlated with hepatic inflammation and contents of MDA and NO, while they were negatively correlated with the activities and the expressions of antioxidant enzymes. The hepatic inflammation was positively correlated with contents of MDA and NO, while it was negatively correlated with the activities and the expressions of antioxidant enzymes. The AMPK activation was negatively correlated with lipid synthesis and inflammation, while it was positively correlated with antioxidant capacity. 

## 3. Discussion

NAFLD has become a major global health problem, which leads to an increased risk of obesity, diabetes mellitus, and cardiovascular disease [3]. However, there have been no approved pharmacological therapies so far [18], and therefore, it is urgent to develop effective therapies for prevention and treatment of NAFLD. Recently, dietary natural compounds have gained a great deal of attention to provide promising strategies for the improvement of NAFLD [19,20]. COS is a natural oligomer that possesses diverse beneficial biological activities, including anti-inflammatory, antioxidant, antimicrobial, antitumor, antiobesity, and anti-hypertensive activities [13]. In the present study, we identified a beneficial effect of COS on prevention of HFD-induced NAFLD. We found that COS treatment, especially at a high dose of 400 mg/kg BW, could reduce serum lipid profile and ameliorate hepatic lipid accumulation, as well as decreasing hepatic inflammation and oxidative stress, so as to alleviate hepatic injury and steatosis in mice. The dose of 400 mg/kg in mouse corresponds to a human-equivalent dose of 32.5 mg/kg, which was calculated according to the report by Nair et al [21]. Our study provides critical understanding of the effect of COS on liver homeostasis regulation. 

Previous studies demonstrated that NAFLD could be induced in rodent models via long-term feeding of HFD [5,22]. In the present study, C57BL/6 mice fed an HFD (5.24 kcal/g, 60% calories from fat) for 7 weeks resulted in increases of serum lipid levels and hepatic lipid deposition. However, administration with COS significantly decreased HFD-induced high levels of serum TC, LDL, and LDL/HDL. Notably, we found the serum HDL level in the HFD-fed mice was slightly increased by COS, and this effect may related to improved lipid catabolism, as HDL can transport cholesterol from peripheral tissues to the liver for reutilization [13]. Importantly, in addition to the decreased liver weight and reduced hepatic TC and TG levels owing to COS administration, the histological examinations further demonstrated the protective effect of COS on hepatic steatosis. The inconsistent effects of COS on TG levels in serum and in liver were observed. COS did not affect the serum TG level, presumably because the serum TG level highly varied with the lipid flux through the liver, adipose tissue, and other tissues [23]. Accumulated evidences have demonstrated that chitosan could decrease plasma and hepatic lipids in HFD-fed animals by promoting faecal fat excretion and directly binding to TG, cholesterol, and bile acids [24]. With the lower molecular weight and the higher water solubility, COS is more readily absorbed through the intestine and distributed to blood and liver than chitosan, and it leads to a better lipid-lowering effect [25]. Moreover, we found that mice fed with HFD showed significantly higher serum levels of AST and ALT compared to the NCD group. AST and ALT are important indices related to liver function, and the abnormal increased serum levels of AST and ALT implied occurrence of liver injury and hepatotoxicity, which are closely associated with hyperlipidemia and hepatic steatosis [26]. COS administration lowers HFD-induced high serum levels of AST and ALT, indicating that COS could protect mice against HFD-induced liver damage.

As a transcription factor, *SREBP-1c* plays a critical role in hepatic fatty acid synthesis by regulating the expression of the enzymes concerning fatty acids formation, such as *FAS* [27,28]. Our results showed that the COS administration downregulated the HFD-induced elevated mRNA expressions of *SREBP-1c* and *FAS* in the liver, which contribute to the ameliorated effect of COS on the hepatic lipid accumulation. PPARα is the major regulator of lipid metabolism, mainly working by activating the fatty acid β-oxidation in liver mitochondria and peroxisome [29]. Previous studies demonstrated that the lack of PPARα aggravated hepatic steatosis in HFD-fed mice [30]. PPARα also regulates the gene expression of *CPT-1*, which is an enzyme located in the mitochondrial membrane to catalyze the transfer of long-chain fatty acids into mitochondria [31]. We found that the mRNA levels of fatty β-oxidation-related genes *PPARα* and *CPT-1* were upregulated by the administration with COS. Taken together, our results suggested that COS might have beneficial effects on HFD-induced hepatic steatosis by inhibiting lipogenesis and promoting fatty acid oxidation.

In NAFLD, inflammatory response may be promoted by various stimuli, such as TC, cholesterol, free fatty acids, and gut-derived endotoxins, and is characterized by increases of pro-inflammatory cytokines in the liver [32]. Conversely, inflammatory and pro-inflammatory molecules may lead to a stress response of hepatocytes and result in hepatic lipid overloading, and thus, they produce direct influence on pathogenesis of NAFLD [33]. Li et al. reported that NAFLD was improved in HFD-fed mice treated with anti-TNF antibodies [34], indicating that the drugs, which can inhibit inflammatory response, may be beneficial for NAFLD. Numerous studies in vitro and in vivo have demonstrated that COS has an anti-inflammatory activity [13]. Bai et al. reported that COS treatment downregulated the mRNA levels of pro-inflammatory cytokines (TNF-α, IL-6, and monocyte chemoattractant protein 1(MCP-1)) both in the palmitic acid-induced HepG2 cells and in the liver of HFD-induced mice [17]. In consistency with these findings, our results of enzyme-linked immunosorbent assay (ELISA) experiments showed that COS administration could inhibit HFD-induced hepatic inflammation by decreasing TNF-α and IL-6 levels. Liver immune homeostasis is mainly regulated by liver-resident macrophages (Kupffer cells) [32]. In NAFLD, the neutrophils are recruited by liver-resident Kupffer cells to the liver [35]. To evaluate the effect of COS on the neutrophil infiltration in the liver, we determined the activity of MPO, an enzyme secreted by neutrophilic granulocytes. The result showed that the elevated hepatic MPO activity induced by HFD was suppressed by COS, indicating a reduction in neutrophil infiltration. In addition, the Kupffer cells also recruited monocytes to the liver, where they were rapidly differentiated and polarized into M1-type macrophages (pro-inflammatory macrophages) in the process of NAFLD [36]. We found the mRNA expressions of *F4/80* and *CD11c*—M1-type macrophages markers—were reduced by COS, which was further supported by the immunohistochemical staining results of F4/80. The improved macrophage polarization was consistent with the ELISA results of pro-inflammatory cytokines, which were secreted by activated resident Kupffer cells and recruited monocyte-derived macrophages [32]. These data suggested that the mechanism, by which COS administration alleviates HFD-induced NAFLD, might partly be attributable to reduced macrophages and neutrophil recruitment, as well as macrophage activation and polarization.

Oxidative stress has often been linked to NAFLD. Matsuzawa et al. reported that the accumulation of TC, cholesterol, and free fatty acids induced by HFD accelerated oxidative stress in the liver [37]. The hepatic steatosis has been demonstrated to correlate negatively with an antioxidant capacity [38], and thus, several natural antioxidants were reported to be effective in prevention and therapy of NAFLD [39]. In this study, HFD-fed mice showed lowered activities of antioxidant enzymes GSH-Px and elevated levels of MDA, an index that reflects the lipid peroxidation degree, as well as overproduction of free radical NO in the liver, suggesting an increased oxidative stress and a decreased antioxidant capacity. However, COS administration enhanced the antioxidant capacity via increases of GSH-Px and SOD activities, and it reduced the hepatic levels of MDA and NO. To thoroughly clarify the potential mechanisms involved in the antioxidative effect of COS, we further examined the protein expression of Nrf2 and mRNA levels of its downstream genes, including *NQO1*, *HO-1*, and *GSTA1*. Nrf2 is well known to be a transcription factor that plays an important role in maintaining liver redox homeostasis, and it improves the defense capacity against oxidative stress by activating the transcription of antioxidant genes [40]. In the present study, we found that COS increased the protein expression of Nrf2 and the mRNA levels of *NQO1*, *HO-1*, and *GSTA1* in the liver. Taken together, these data suggested that COS might have a protective effect against HFD-induced oxidative stress by activating the Nrf2 pathway and upregulating genes of antioxidant enzymes. Notably, the free radical NO also can act as a second messenger to participate in the signal transduction and thus caused inflammatory response [41]. Additionally, activation of the Nrf2 pathways has been demonstrated to be involved in suppressing inflammatory response, suggesting that hepatic oxidative stress significantly correlated with inflammation in NAFLD [38,42]. Consistently, we found the Nrf2 activation was negatively correlated with the mRNA and protein expressions of F4/80. Therefore, the anti-inflammatory and antioxidant actions of COS may contribute to its beneficial effect on NAFLD via activation of the Nrf2 pathways.

AMPK is a key regulator to maintain cellular energy homeostasis and regulate diverse metabolic processes [43]. It is well known to inhibit lipid synthesis and stimulate fatty acid oxidation by inducing ACC phosphorylation, suggesting that the AMPK signaling pathway is a potential target for prevention and treatment of NAFLD [44]. In the present study, we found that COS reduced the hepatic fat accumulation by increasing the levels of phosphorylated AMPK and ACC in HFD-fed mice. Muanprasat et al. reported that the mechanisms of COS-induced AMPK activation in intestinal epithelial cells involved the calcium-sensing receptor (CaSR)-mediated calcium release from the endoplasmic reticulum via the phospholipase C (PLC)–IP_3_ receptor channel [45], but the mechanism in liver requires further research. Interestingly, several studies in vitro and in vivo indicated that the activation of AMPK is associated with decreased inflammation [43]. Moreover, recent studies have indicated that AMPK could stimulate the Nrf2 signaling pathway to induce antioxidant defense [46,47]. These finding would be consistent with our correlation analysis results that AMPK activation was negatively correlated with inflammation, while it was positively correlated with antioxidant capacity. Therefore, the AMPK pathway may be involved in the anti-inflammatory and antioxidant actions of COS, which contribute to preventing NAFLD.

## 4. Materials and Methods 

### 4.1. Materials

COS was provided by Panan Yinzhou Biological Products Company (Zhejiang, China). The average molecular weight of COS was 5000 Da. The polymerization degrees were 2–6, and the percentages of these oligomers were 4.66%, 14.28%, 27.71%, 29.27%, and 18.01%, respectively. Primary antibodies against AMPKα and p-AMPKα were purchased from Cell Signaling Technology (Beverly, MA, USA). Anti-ACC, anti-Nrf2, and anti-β-actin were supplied by Proteintech Group, Inc (Wuhan, China). Anti-p-ACC was purchased from Abcam (Cambridge, England). Anti-F4/80 was supplied by Servicebio Technology Co., Ltd. (Wuhan, China).

### 4.2. Animals and Experimental Design

Thirty-two male C57BL/6 mice (4-week-old, Slaccas Experimental Animal Co., Ltd., Shanghai, China) were housed in a controlled room (temperature: 22 ± 2 ℃, relative humidity: 50% ± 10%) with a 12 h light/dark cycle. All experimental procedures were approved by the Animal Care and Use Committee of Zhejiang University (ZJU20190090). After a 1-week adaption period, the mice were randomly divided into four groups with 8 mice in each group as follows: (1) normal chow diet (NCD) group, (2) high-fat diet (HFD) group, (3) HFD + low dosage of COS (200 mg/kg BW) (HFDLC) group, and (4) HFD + high dosage of COS (400 mg/kg BW) (HFDHC) group. The NCD (4% kcal% fat, #P1101F) was provided by Shanghai Puluteng Biological Technology Co., Ltd. (Shanghai, China), and the HFD (60% kcal% fat, #MD12033) was purchased from Jiangsu Medicience Co., Ltd. (Yangzhou, China). The mice in the HFDLC and HFDHC groups received different doses of COS by intragastric administration once a day for 7 weeks, while the mice in the NCD and HFD groups received the same amount of PBS. The doses of COS were selected based on our preliminary experimental result and previous reports [17,48]. The mice were given free access to food and water. At the end of the feeding trial, all mice were anesthetized, and blood samples were collected through retro-orbital plexus and separated by centrifugation at 1000 *g* for 10 min at 4 °C to collect serum samples. Subsequently, all the mice were euthanized, and the liver were harvested, weighed and immediately stored at −80 °C.

### 4.3. Serum Biochemical Analyses

Serum levels of TC, TG, HDL, LDL, AST, and ALT were measured with commercial assay kits (Nanjing Jiancheng Biomedical Company, Nanjing, China).

### 4.4. Hepatic Biochemical Assays

The liver samples (100 mg per mice) were homogenized with 0.9 mL of cold normal saline (0.9%) with a homogenizer and then centrifuged at 12,000× *g* for 15 min at 4 °C. The supernatant was collected for the determination of TC, TG, MPO, T-AOC, GSH-Px, SOD, MDA, and CAT using commercial assay kits (Nanjing Jiancheng Biomedical Company, Nanjing, China). NO was measured with commercial assay kits from Beyotime Biotechnology (Shanghai, China). 

### 4.5. Histological Staining

The liver samples were fixed in a 4% paraformaldehyde solution. After fixing, the tissues were embedded in paraffin, cut into slices of 5 μm thickness and stained with H&E. For Oil Red O staining, the liver tissue was frozen-fixed in an optimal cutting temperature compound (Sakura, Torrance, CA, USA), and sections of 10 μm thickness were cut and stained with Oil Red O to detect lipid droplets. The samples were observed and photographed using a light microscope.

### 4.6. Immunohistochemistry and ELISA

The paraffin sections (5 μm thickness) of livers were dewaxed and hydrated, and they were then incubated with primary F4/80 antibody for 1 h, followed by incubation with secondary antibodies for 30 min. The integrated optical density (IOD)/area ratio of the positive staining in each section was analyzed with Image-Pro Plus 6.0 software (Media Cybernetics, Inc., Rockville, MD, USA).

Pro-inflammatory cytokine levels in the liver homogenates were quantified using ELISA kits specified for mouse TNF-α, IL-1β, and IL-6 (MLBIO Biotechology, Shanghai, China) according to the manufacturer’s instructions.

### 4.7. Quantitative Real-Time PCR

The total RNA was extracted from the liver tissue using Trizol Reagent (Invitrogen, Carlsbad, CA, USA), and cDNA was synthesized from RNA (1 μg) using the reverse transcription reagent kit (TaKaRa, Dalian, China). Real-time PCR was performed with SYBR Green master mixes (TaKaRa, Dalian, China) using a CFX96 Real-Time PCR system (Bio-Rad, Hercules, CA, USA) with the thermal cycle conditions: 1 cycle at 95 °C for 30 s; 40 cycles at 95 °C for 5 s and at 61 °C for 35 s. The primer sequences are listed in Table 2, and the mRNA expressions of the target genes were normalized to that of β-actin and calculated using the 2^−△△CT^ method.

### 4.8. Western Blot Analysis

Total proteins were extracted from the liver tissues with a cold RIPA buffer (Beyotime Biotechnology, Shanghai, China), followed by centrifugation at 12,000 *g* for 10 min at 4 °C. The protein concentrations were determined using a BCA assay kit (KeyGen BioTech, Nanjing, China). Equal amounts of protein were electrophoresed by SDS-PAGE and then transferred onto PVDF membranes (Millipore Corp., Bedford, MA, USA). After blocking with 5% skim milk powder, the membranes were incubated with primary antibodies at 4 °C overnight, followed by secondary antibodies for 1 h at room temperature. Subsequently, the protein bands were visualized using enhanced chemiluminescence (ECL) detection kits (Beyotime Biotechnology, Shanghai, China). ImageJ software was used to quantify the optical density of protein bands. 

### 4.9. Statistical Analysis

All data are expressed as mean ± SE and were analyzed using SPSS 20.0 software (SPSS, Chicago, IL, USA). Statistical analysis was performed using one-way ANOVA followed by Duncan’s multiple-comparison test. Spearman’s correlation coefficients were analyzed to define pair-wise associations between the studied variables using GraphPad Prism 7.0 (San Diego, CA, USA). Differences with *p* < 0.05 were considered statistically significant.

## 5. Conclusions

In conclusion, the present study demonstrated that COS exerted protective effects against NAFLD in HFD-fed C57BL/6 mice. We further found COS activated the AMPK signaling pathway to regulate lipid metabolism and inhibit inflammation response and oxidative stress to prevent hepatic damage. The observations suggest that COS might be an effective dietary supplement to ameliorate NAFLD and related metabolic diseases.

## Figures and Tables

**Figure 1 marinedrugs-17-00645-f001:**
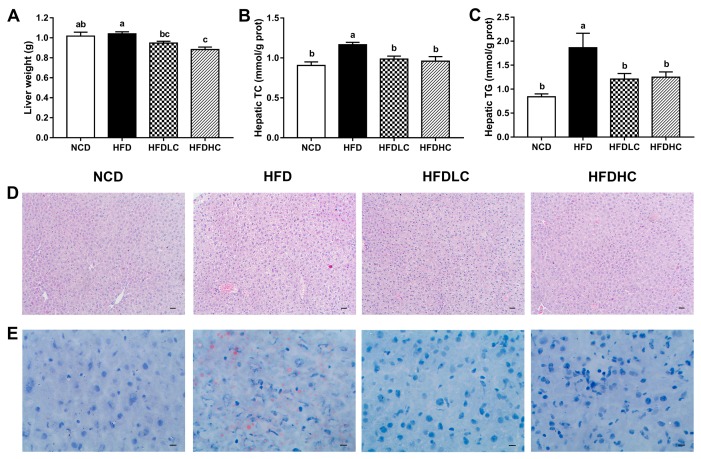
Effects of COS on hepatic steatosis in mice: (**A**) liver weight; (**B**) hepatic total cholesterol (TC); (**C**) hepatic triacylglycerol (TG); (**D**) hematoxylin and eosin (H&E) staining of liver sections (scale bar, 30 μm); (**E**) Oil Red O staining of liver sections (scale bar, 9 μm). Data are presented as the mean ± SE (*n* = 8 per group). Results were statistically analyzed using one-way ANOVA followed by Duncan’s multiple-comparison test, and values with different labels (a–c) are significantly different (*p* < 0.05). NCD, normal chow diet group; HFD, high-fat diet group; HFDLC, high-fat diet + low dosage of COS (200 mg/kg BW) group; HFDHC, high-fat diet + high dosage of COS (400 mg/kg BW) group; TC, total cholesterol; TG, triacylglycerol.

**Figure 2 marinedrugs-17-00645-f002:**
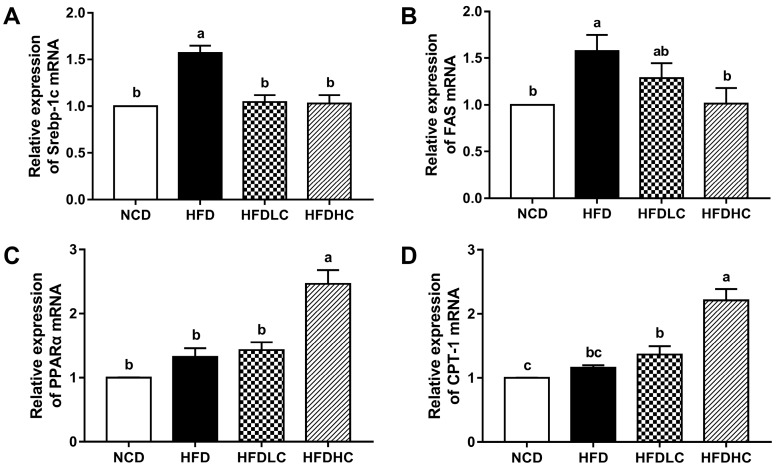
Effects of COS on the mRNA expression of genes related to hepatic lipid metabolism in mice: (**A**) *SREBP-1c*; (**B**) *FAS*; (**C**) *PPARα*; (**D**) *CPT-1*. Data are presented as the mean ± SE (*n* = 6 per group). Results were statistically analyzed using one-way ANOVA followed by Duncan’s multiple-comparison test, and values with different labels (a–c) are significantly different (*p* < 0.05). NCD, normal chow diet group; HFD, high-fat diet group; HFDLC, high-fat diet + low dosage of COS (200 mg/kg BW) group; HFDHC, high-fat diet + high dosage of COS (400 mg/kg BW) group. SREBP-1c, sterol regulatory element-binding protein-1c; FAS, fatty acid synthase; PPARα, peroxisome proliferator-activated receptor alpha; CPT-1, carnitine palmitoyltransferase 1.

**Figure 3 marinedrugs-17-00645-f003:**
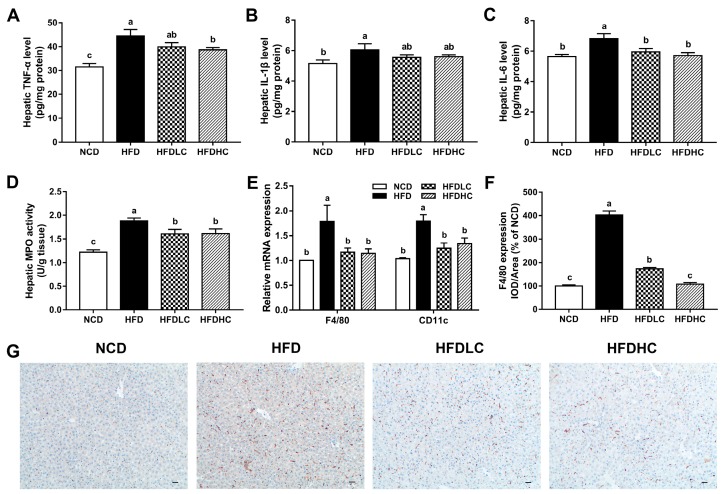
Effects of COS on hepatic inflammation in mice: (**A**) hepatic TNF-α level; (**B**) hepatic IL-1β level; (**C**) hepatic IL-6 level; (**D**) hepatic MPO activity; (**E**) relative mRNA expression of *F4/80* and *CD11c*; (**F**) the IOD/area ratio of F4/80; (**G**) immunohistochemical staining of F4/80 in the liver (scale bar, 30 μm). Data are presented as the mean ± SE (*n* = 6 per group). Results were statistically analyzed using one-way ANOVA followed by Duncan’s multiple-comparison test, and values with different labels (a–c) are significantly different (*p* < 0.05). NCD, normal chow diet group; HFD, high-fat diet group; HFDLC, high-fat diet + low dosage of COS (200 mg/kg BW) group; HFDHC, high-fat diet + high dosage of COS (400 mg/kg BW) group. TNF-α, tumor necrosis factor-α; IL-1β, interleukin-1β; IL-6, interleukin-6; MPO, myeloperoxidase; IOD, integrated optical density.

**Figure 4 marinedrugs-17-00645-f004:**
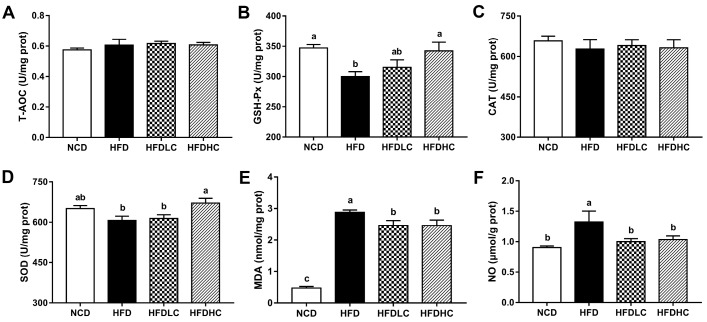
Effects of COS on hepatic antioxidant indicators in mice: activities of T-AOC (**A**), GSH-Px (**B**), CAT (**C**), and SOD (**D**) in the liver; levels of MDA (**E**) and NO (**F**) in the liver. Data are presented as the mean ± SE (*n* = 8 per group). Results were statistically analyzed using one-way ANOVA followed by Duncan’s multiple-comparison test, and values with different labels (a–c) are significantly different (*p* < 0.05). NCD, normal chow diet group; HFD, high-fat diet group; HFDLC, high-fat diet + low dosage of COS (200 mg/kg BW) group; HFDHC, high-fat diet + high dosage of COS (400 mg/kg BW) group. T-AOC, total antioxidant capacity; GSH-Px, glutathione peroxidase; CAT, catalase; SOD, superoxide dismutase; MDA, malondialdehyde; NO, nitric oxide.

**Figure 5 marinedrugs-17-00645-f005:**
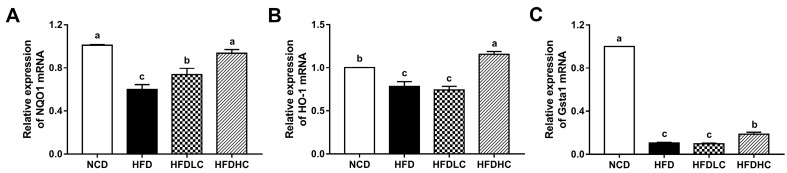
Effects of COS on mRNA expression related to hepatic antioxidant capacity in mice: relative mRNA expressions of *NQO1* (**A**), *HO-1* (**B**), and *GSTA1* (**C**) in livers. Data are presented as the mean ± SE (*n* = 5−6 per group). Results were statistically analyzed using one-way ANOVA followed by Duncan’s multiple-comparison test, and values with different labels (a–c) are significantly different (*p* < 0.05). NCD, normal chow diet group; HFD, high-fat diet group; HFDLC, high-fat diet + low dosage of COS (200 mg/kg BW) group; HFDHC, high-fat diet + high dosage of COS (400 mg/kg BW) group. NQO1, NADPH quinone oxidoreductase 1; HO-1, heme oxygenase-1; GSTA1, glutathione S-transferase alpha 1.

**Figure 6 marinedrugs-17-00645-f006:**
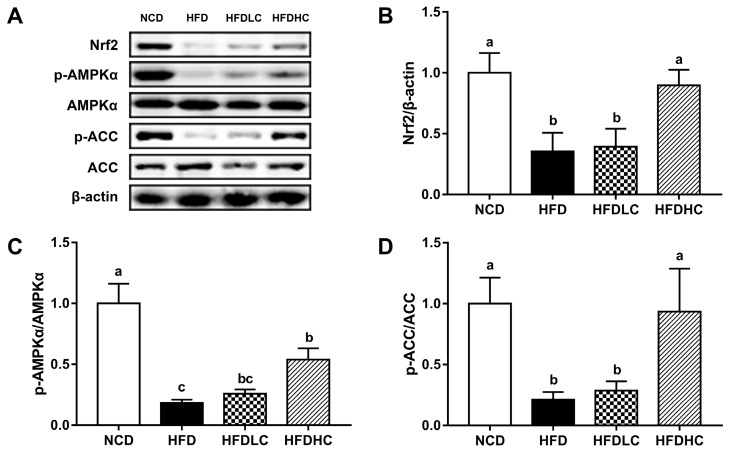
Effect of COS on hepatic protein expression in mice: (**A**) protein expressions of Nrf2, p-AMPK, AMPK, p-ACC, and ACC analyzed by the Western blot; density identification of Nrf2 (**B**), AMPK (**C**), and ACC (**D**). Data presented as the mean ± SE (*n* = 4−6 per group). Results were statistically analyzed using one-way ANOVA followed by Duncan’s multiple-comparison test, and values with different labels (a–c) are significantly different (*p* < 0.05). NCD, normal chow diet group; HFD, high-fat diet group; HFDLC, high-fat diet + low dosage of COS (200 mg/kg BW) group; HFDHC, high-fat diet + high dosage of COS (400 mg/kg BW) group. Nrf2, nuclear factor E2-related factor 2; AMPK, adenosine monophosphate-activated protein kinase; ACC, acetyl-CoA carboxylase.

**Figure 7 marinedrugs-17-00645-f007:**
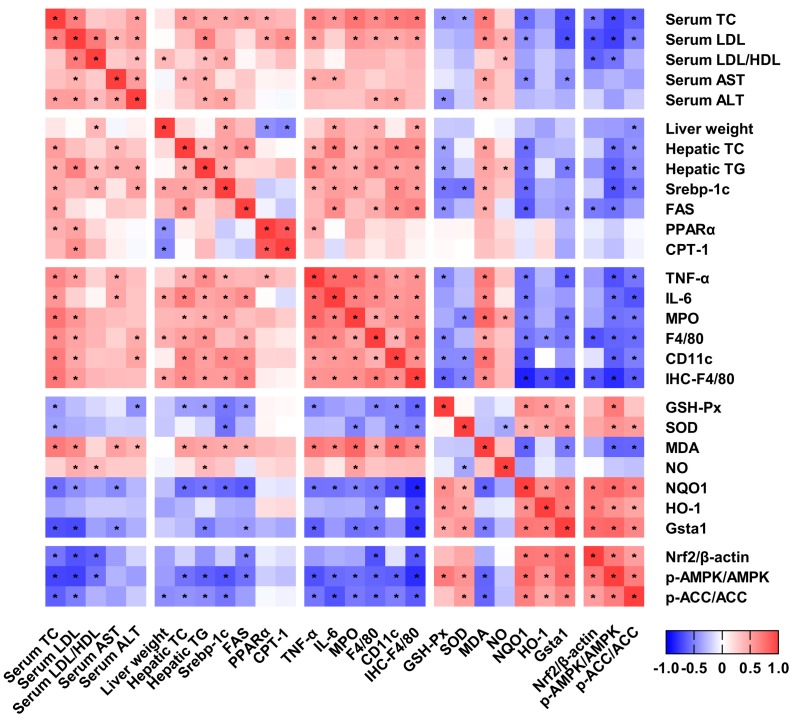
Correlation between the studied variables. Spearman’s correlation coefficients were analyzed by GraphPad Prism 7.0 (San Diego, CA, USA), and cells were marked with asterisk, indicating two variables were significantly associated with each other (*p* < 0.05).

**Table 1 marinedrugs-17-00645-t001:** Effects of chitosan oligosaccharide (COS) on serum biochemical parameters in mice.

	NCD	HFD	HFDLC	HFDHC
TC (mmol/L)	3.03 ± 0.068 ^c^	5.88 ± 0.295 ^a^	5.54 ± 0.307 ^a^	4.82 ± 0.151 ^b^
TG (mmol/L)	0.44 ± 0.020	0.48 ± 0.026	0.45 ± 0.018	0.44 ± 0.025
HDL (mmol/L)	3.48 ± 0.234 ^b^	6.01 ± 0.250 ^a^	6.48 ± 0.188 ^a^	6.43 ± 0.294 ^a^
LDL (mmol/L)	0.53 ± 0.032 ^c^	1.28 ± 0.142 ^a^	1.01 ± 0.112 ^a,b^	0.98 ± 0.044 ^b^
LDL/HDL	0.154 ± 0.010 ^b^	0.216 ± 0.027 ^a^	0.155 ± 0.015 ^b^	0.154 ± 0.007 ^b^
AST (U/L)	14.06 ± 1.234 ^b^	21.36 ± 2.668 ^a^	17.61 ± 1.095 ^a,b^	14.89 ± 0.810 ^b^
ALT (U/L)	7.55 ± 0.688 ^b^	9.98 ± 0.438 ^a^	8.41 ± 0.559 ^a,b^	7.81 ± 0.555 ^b^

Data are presented as the mean ± SE (*n* = 8 per group). Results were statistically analyzed using one-way ANOVA followed by Duncan’s multiple-comparison test, and values with different labels (a–c) within each row are significantly different (*p* < 0.05). NCD, normal chow diet group; HFD, high-fat diet group; HFDLC, high-fat diet + low dosage of COS (200 mg/kg BW) group; HFDHC, high-fat diet + high dosage of COS (400 mg/kg BW) group. TC, total cholesterol; TG, triacylglycerol; HDL, high-density lipoprotein; LDL, low-density lipoprotein; AST, aspartate aminotransferase; ALT, alanine aminotransferase.

**Table 2 marinedrugs-17-00645-t002:** Primer sequences used in real-time PCR.

Gene	Forward primer (5’−3’)	Reverse primer (5’−3’)
*SREBP-1c*	GCCATCGACTACATCCGCTTCTTG	TGCCTCCTCCACTGCCACAAG
*FAS*	GGAGGTGGTGATAGCCGGTAT	TGGGTAATCCATAGAGCCCAG
*PPARα*	CAGGAGAGCAGGGATTTGCA	CCTACGCTCAGCCCTCTTCAT
*CPT-1*	ATGGCAGAGGCTCACCAAGC	GATGAACTTCCAGGAGTGC
*F4/80*	TGGCAAGCATCATGGCATACCTG	TGACGGTTGAGCAGACAGTGAATG
*CD11c*	AGACGTGCCAGTCAGCATCAAC	GCAGTCAGCGATGGAGCAGTC
*NQO1*	AGGCTGCTGTAGAGGCTCTGAAG	GCTCAGGCGTCCTTCCTTATATGC
*HO-1*	GAGCAGAACCAGCCTGAACTA	GGTACAAGGAAGCCATCACCA
*GSTA1*	TGCCCAATCATTTCAGTCAG	CCAGAGCCATTCTCAACTA
*β-actin*	CGTTGACATCCGTAAAGACC	AACAGTCCGCCTAGAAGCAC

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
