# Peer review of "Chitosan Oligosaccharide Attenuates Nonalcoholic Fatty Liver Disease Induced by High Fat Diet through Reducing Lipid Accumulation, Inflammation and Oxidative Stress in C57BL/6 Mice"

_marinedrugs, 2019, doi:10.3390/md17110645_

Round 1
Reviewer 1 Report
The Authors report a positive effect of chitosan oligosaccharide (COS) in NAFLD disease induced by high fat diet in C57BL/6 mice
Usually in human chitosan is used to reduce body weight reducing the absorbance of fat and the dose is about 40 mg /Kg BW and this point should be considered in the discussion
I would mitigate the explanation of the pathogenesis of NAFLD using this experimental model. It is well known that in humans the involved factors are multiple including genetic, insulin resistance, individual metabolism and diet. The role of inflammation in obesity and NAFLD is very important but also other factors are involved also in the inflammation.
From these results is seems that NAFLD is mostly due to HFD and the introduction of a NCD diet would be adequate to prevent the disease. COS can decrease the values of parameters of HFD but their values are very high compared with NCD. The Authors should also report possible side effects during COS treatment.
In the table 1 the effect of COS using different chow diet is reported.
The authors should explain the criteria to establish that 200 mg/Kg of BW is a low dosage and 400 mg is a high dosage, considering that the concentration of chitosan in humans is about 40 mg/Kg
The table 1 it is not clear and the authors should explain the significance a b c
The same table shows a clear effect of COS in ameliorating the TC HDL and LDL in mice
From the results it seems that HFD induces a sharp increase oh HDL cholesterol from 3.48 to 6.1, but the effect of COS in decreasing TC is only evident with HFDHC. It is important to note that the amount of COS used in this experimental model is high compared to those used in humans.
Page 4 it is not clear if cytokines and TNF alpha are produced by the liver or by local infiltrating inflammatory cells (lymphocytes macrophages.)
Line 224 chitosan could decrease plasma and hepatic lipids: in table 1 TC is not reduced in HFDLC and in particular HDL cholesterol is increased
In conclusion the study is of interest but the authors should discuss about the different amounts of COS used in these mice and in humans to be sure that the dose used in mice can be used also in human NAFLD.
Reviewer 2 Report
This study was designed to examine the effect of chitosan oligosaccharide (COS) on lipid accumulation, inflammation, and oxidative stress in HFD-fed mice. The study is well conducted and there are very interesting data in this manuscript. However, the reviewer has the minor concerns prior to publication;
- Please indicate the body weights and food intake during COS treatments. Additionally, it is better to show the relative liver weights (g/kg BW) in Figure 1.
- The authors should provide the explanation for the selection of doses of 200 and 400 mg/kg COS in this study. Additionally, will these doses be achievable in humans?
- In Table 1, there was no significantly difference in the serum TG levels between the NCD and HFD groups. However, the hepatic TG level in the HFD group were higher than that in the NCD group (Figure 1). The authors should provide the reason in the Discussion section.
- Why was the AMPK activity upregulated by the COS treatment? The authors should provide its mechanism in the Discussion section.
Round 2
Reviewer 1 Report
The authors have answered to the comments and I don't have further quesdtions